# Training Deep Models Faster with Robust, Approximate Importance Sampling

**Tyler B. Johnson**
University of Washington, Seattle
tbjohns@washington.edu

**Carlos Guestrin**
University of Washington, Seattle
guestrin@cs.washington.edu

## Abstract

In theory, importance sampling speeds up stochastic gradient algorithms for supervised learning by prioritizing training examples. In practice, the cost of computing importances greatly limits the impact of importance sampling. We propose a robust, approximate importance sampling procedure (RAIS) for stochastic gradient descent. By approximating the ideal sampling distribution using robust optimization, RAIS provides much of the benefit of exact importance sampling with drastically reduced overhead. Empirically, we find RAIS-SGD and standard SGD follow similar learning curves, but RAIS moves faster through these paths, achieving speed-ups of at least 20% and sometimes much more.

## 1 Introduction

Deep learning models perform excellently on many tasks. Training such models is resource-intensive, however, as stochastic gradient descent algorithms can require days or weeks to train effectively. After a short period training, models usually perform well on some—or even most—training examples. As training continues, frequently reconsidering such "easy" examples slows further improvement.

Importance sampling prioritizes training examples for SGD in a principled way. The technique suggests sampling example $i$ with probability proportional to the norm of loss term $i$'s gradient. This distribution both prioritizes challenging examples and minimizes the stochastic gradient's variance.

SGD with optimal importance sampling is impractical, however, since computing the sampling distribution requires excessive time. [1] and [2] analyze importance sampling for SGD and convex problems; practical versions of these algorithms sample proportional to fixed constants. For deep models, other algorithms attempt closer approximations of gradient norms [3, 4, 5]. But these algorithms are not inherently robust. Without carefully chosen hyperparameters or additional forward passes, these algorithms do not converge, let alone speed up training.

We propose RAIS, an importance sampling procedure for SGD with several appealing qualities. First, RAIS determines each sampling distribution by solving a robust optimization problem. As a result, each sampling distribution is minimax optimal with respect to an uncertainty set. Since RAIS trains this uncertainty set in an adaptive manner, RAIS is not sensitive to hyperparameters.

In addition, RAIS maximizes the benefit of importance sampling by adaptively increasing SGD's learning rate—an effective yet novel idea to our knowledge. This improvement invites the idea that one RAIS-SGD iteration equates to more than one iteration of conventional SGD. Interestingly, when plotted in terms of "epochs equivalent," the learning curves of the algorithms align closely.

RAIS applies to any model that is trainable with SGD. RAIS also combines nicely with standard "tricks," including data augmentation, dropout, and batch normalization. We show this empirically in §6. In this section, we also demonstrate that RAIS consistently improves training times. To provide context for the paper, we include qualitative results from these experiments in Figure 1.

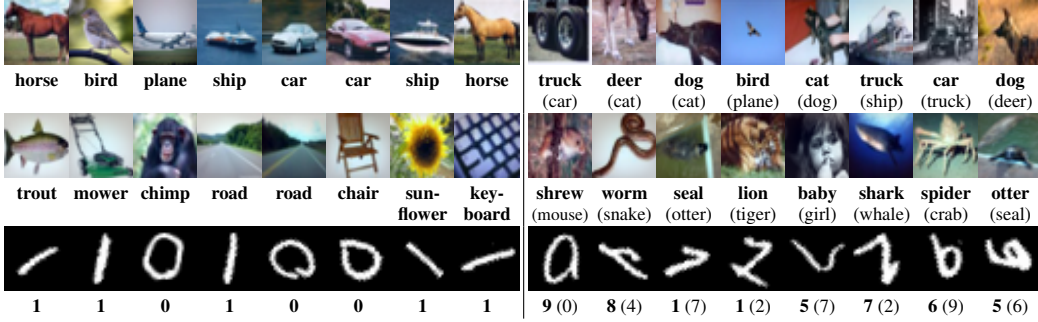

**Figure 1: Nonpriority and priority training examples for image classification.** *Left:* Examples that RAIS samples infrequently during training. *Right:* Examples that RAIS prioritizes. Bold denotes the image's label. Parentheses denote a different class that the model considers likely during training. Datasets are CIFAR-10 (top), CIFAR-100 (middle), and rotated MNIST (bottom).

## 2 Problem formulation

Given loss functions $f_1, f_2, \ldots, f_n$ and a tuning parameter $\lambda \in \mathbb{R}_{\geq 0}$, our task is to efficiently solve

$$\underset{\mathbf{w} \in \mathbb{R}^d}{\text{minimize}} \; F(\mathbf{w}), \quad \text{where} \quad F(\mathbf{w}) = \frac{1}{n} \sum_{i=1}^{n} f_i(\mathbf{w}) + \frac{\lambda}{2} \|\mathbf{w}\|^2 . \tag{P}$$

A standard algorithm for solving (P) is stochastic gradient descent. Let $\mathbf{w}^{(t)}$ denote the optimization variables when iteration $t$ begins. SGD updates these weights via

$$\mathbf{w}^{(t+1)} \leftarrow \mathbf{w}^{(t)} - \eta^{(t)} \mathbf{g}^{(t)} . \tag{1}$$

Above, $\eta^{(t)} \in \mathbb{R}_{>0}$ is a learning rate, specified by a schedule: $\eta^{(t)} = \texttt{lr\_sched}\,(t)$. The vector $\mathbf{g}^{(t)}$ is an unbiased stochastic approximation of the gradient $\nabla F(\mathbf{w}^{(t)})$. SGD computes $\mathbf{g}^{(t)}$ by sampling a minibatch of $|\mathcal{M}|$ indices from $\{1, 2, \ldots, n\}$ uniformly at random (or approximately so). Denoting this minibatch by $\mathcal{M}^{(t)}$, SGD defines the stochastic gradient as

$$\mathbf{g}^{(t)} = \frac{1}{|\mathcal{M}|} \sum_{i \in \mathcal{M}^{(t)}} \nabla f_i(\mathbf{w}^{(t)}) + \lambda \mathbf{w}^{(t)} . \tag{2}$$

In this work, we assume an objective function, learning rate schedule, and minibatch size, and we propose a modified algorithm called RAIS-SGD. RAIS prioritizes examples by sampling minibatches *non-uniformly*, allowing us to train models using fewer iterations and less time.

## 3 SGD with oracle importance sampling

We now introduce an SGD algorithm with "oracle" importance sampling, which prioritizes examples using exact knowledge of importance values. RAIS-SGD is an approximation of this algorithm.

Given $\mathbf{w}^{(t)}$, let us define the expected training progress attributable to iteration $t$ as

$$\mathbb{E}\Delta^{(t)} = \|\mathbf{w}^{(t)} - \mathbf{w}^\star\|^2 - \mathbb{E}\left[\|\mathbf{w}^{(t+1)} - \mathbf{w}^\star\|^2\right]$$
$$= 2\eta^{(t)} \langle \nabla F(\mathbf{w}^{(t)}), \mathbf{w}^{(t)} - \mathbf{w}^\star \rangle - [\eta^{(t)}]^2 \mathbb{E}\left[\|\mathbf{g}^{(t)}\|^2\right] . \tag{3}$$

Here $\mathbf{w}^\star$ denotes the solution to (P), and the expectation is with respect to minibatch $\mathcal{M}^{(t)}$. The equality follows from plugging in (1) and applying the fact that $\mathbf{g}^{(t)}$ is unbiased.

We refer to our oracle algorithm as O-SGD, and we refer to SGD with uniform sampling as U-SGD. At a high level, O-SGD makes two changes to U-SGD in order to increase $\mathbb{E}\Delta^{(t)}$. First, O-SGD samples training examples non-uniformly in a way that minimizes the variance of the stochastic gradient. This first change is not new—see [1], for example. Second, to compensate for the first improvement, O-SGD adaptively increases the learning rate. This second change, which is novel to our knowledge, can be essential for obtaining large speed-ups.

## 3.1 Maximizing progress with oracle importance sampling

By sampling minibatches non-uniformly, O-SGD prioritizes training examples in order to decrease $\mathbb{E}[\|\mathbf{g}_{\mathrm{O}}^{(t)}\|^2]$. During iteration $t$, O-SGD defines a discrete distribution $\mathbf{p}^{(t)} \in \mathbb{R}_{\geq 0}^n$, where $\sum_i p_i^{(t)} = 1$. O-SGD constructs minibatch $\mathcal{M}^{(t)}$ by sampling independently $|\mathcal{M}|$ examples according to $\mathbf{p}^{(t)}$. Instead of (2), the resulting stochastic gradient is

$$\mathbf{g}_{\mathrm{O}}^{(t)} = \tfrac{1}{|\mathcal{M}|} \sum_{i \in \mathcal{M}^{(t)}} \tfrac{1}{np_i^{(t)}} \nabla f_i(\mathbf{w}^{(t)}) + \lambda \mathbf{w}^{(t)} \,. \tag{4}$$

Scaling the $\nabla f_i$ terms by $(np_i^{(t)})^{-1}$ ensures $\mathbf{g}_{\mathrm{O}}^{(t)}$ remains an unbiased approximation of $\nabla F(\mathbf{w}^{(t)})$. O-SGD defines $\mathbf{p}^{(t)}$ as the sampling distribution that maximizes (3):

**Proposition 3.1** (Oracle sampling distribution). *In order to minimize $\mathbb{E}[\|\mathbf{g}_{\mathrm{O}}^{(t)}\|^2]$, O-SGD samples each example $i$ with probability proportional to the $i$th "gradient norm." That is,*

$$p_i^{(t)} = \|\nabla f_i(\mathbf{w}^{(t)})\| \big/ \sum_{j=1}^n \|\nabla f_j(\mathbf{w}^{(t)})\| \,.$$

*Proof sketch.* Defining $\bar{f}(\mathbf{w}) = \frac{1}{n} \sum_{i=1}^n f_i(\mathbf{w})$, we write this second moment as

$$\mathbb{E}\left[\|\mathbf{g}_{\mathrm{O}}^{(t)}\|^2\right] = \tfrac{1}{n^2|\mathcal{M}|} \sum_{i=1}^n \tfrac{1}{p_i^{(t)}} \|\nabla f_i(\mathbf{w}^{(t)})\|^2 - \tfrac{1}{|\mathcal{M}|} \|\nabla \bar{f}(\mathbf{w}^{(t)})\|^2 + \|\nabla F(\mathbf{w}^{(t)})\|^2 \,. \tag{5}$$

Finding the distribution $\mathbf{p}^{(t)}$ that minimizes (5) is a problem with a closed-form solution. The solution is the distribution defined by Proposition 3.1, which we show in Appendix A. $\qquad\square$

The oracle sampling distribution is quite intuitive. Training examples with largest gradient norm are most important for further decreasing $F$, and these examples receive priority. Examples that the model handles correctly have smaller gradient norm, and O-SGD deprioritizes these examples.

## 3.2 Adapting the learning rate

Because importance sampling reduces the stochastic gradient's variance—possibly by a large amount—we find it important to adaptively increase O-SGD's learning rate compared to U-SGD. For O-SGD, we propose a learning rate that depends on the "gain ratio" $r_{\mathrm{O}}^{(t)} \in \mathbb{R}_{\geq 1}$:

$$r_{\mathrm{O}}^{(t)} = \mathbb{E}\left[\|\mathbf{g}_{\mathrm{U}}^{(t)}\|^2\right] \Big/ \mathbb{E}\left[\|\mathbf{g}_{\mathrm{O}}^{(t)}\|^2\right] \,. \tag{6}$$

Above, $\mathbf{g}_{\mathrm{U}}^{(t)}$ is the stochastic gradient defined by uniform sampling. O-SGD adapts the learning rate so that according to (3), one O-SGD iteration results in as much progress as $r_{\mathrm{O}}^{(t)}$ U-SGD iterations. Defining the edge case $r_{\mathrm{O}}^{(0)} = 1$, this learning rate depends on the "effective iteration number"

$$\hat{t}_{\mathrm{O}}^{(t)} = \sum_{t'=1}^t r_{\mathrm{O}}^{(t'-1)} \,.$$

Since the gain ratio exceeds 1, we have $\hat{t}_{\mathrm{O}}^{(t)} \geq t$ for all $t$. O-SGD defines the learning rate as

$$\eta_{\mathrm{O}}^{(t)} = r_{\mathrm{O}}^{(t)} \mathtt{lr\_sched}(\hat{t}_{\mathrm{O}}^{(t)}) \,.$$

We justify this choice of learning rate schedule with the following proposition:

**Proposition 3.2** (Equivalence of gain ratio and expected speed-up). *Given $\mathbf{w}^{(t)}$, define $\mathbb{E}\Delta_{\mathrm{U}}^{(t)}$ as the expected progress from iteration $t$ of U-SGD with learning rate $\eta_{\mathrm{U}}^{(t)} = \mathtt{lr\_sched}(t)$. For comparison, define $\mathbb{E}\Delta_{\mathrm{O}}^{(t)}$ as the expected progress from iteration $t$ of O-SGD with learning rate $\eta_{\mathrm{O}}^{(t)} = r_{\mathrm{O}}^{(t)} \eta_{\mathrm{U}}^{(t)}$. Then $\mathbb{E}\Delta_{\mathrm{O}}^{(t)} = r_{\mathrm{O}}^{(t)} \mathbb{E}\Delta_{\mathrm{U}}^{(t)}$. Relative to U-SGD, O-SGD multiplies the expected progress by $r_{\mathrm{O}}^{(t)}$.*

*Proof.* Using (3), we have

$$\mathbb{E}\Delta_{\mathrm{U}}^{(t)} = 2\eta_{\mathrm{U}}^{(t)} \langle \nabla F(\mathbf{w}^{(t)}), \mathbf{w}^{(t)} - \mathbf{w}^\star \rangle - [\eta_{\mathrm{U}}^{(t)}]^2 \mathbb{E}\left[\|\mathbf{g}_{\mathrm{U}}^{(t)}\|^2\right] \,.$$

For O-SGD, we expect progress

$$\mathbb{E}\Delta_{\mathrm{O}}^{(t)} = 2\eta_{\mathrm{O}}^{(t)}\langle\nabla F(\mathbf{w}^{(t)}), \mathbf{w}^{(t)} - \mathbf{w}^{\star}\rangle - [\eta_{\mathrm{O}}^{(t)}]^2\mathbb{E}\left[\|\mathbf{g}_{\mathrm{O}}^{(t)}\|^2\right]$$

$$= 2r_{\mathrm{O}}^{(t)}\eta_{\mathrm{U}}^{(t)}\langle\nabla F(\mathbf{w}^{(t)}), \mathbf{w}^{(t)} - \mathbf{w}^{\star}\rangle - r_{\mathrm{O}}^{(t)}[\eta_{\mathrm{U}}^{(t)}]^2\mathbb{E}\left[\|\mathbf{g}_{\mathrm{U}}^{(t)}\|^2\right] = r_{\mathrm{O}}^{(t)}\mathbb{E}\Delta_{\mathrm{U}}^{(t)}\,.$$

$\square$

We remark that the purpose of this learning rate adjustment is not necessarily to speed up training—whether the adjustment results in speed-up depends greatly on the original learning rate schedule. Instead, the purpose of this rescaling is to make O-SGD (and hence RAIS-SGD) suitable as a drop-in replacement for U-SGD. We show empirically that this is the case in §6.

# 4   Robust approximate importance sampling (RAIS)

Determining $\mathbf{p}^{(t)}$ and $r_{\mathrm{O}}^{(t)}$ in O-SGD depends on knowledge of many gradient norms ($\|\nabla f_i(\mathbf{w}^{(t)})\|$ for all examples, $\|\nabla\bar{f}(\mathbf{w}^{(t)})\|$, and $\|\nabla F(\mathbf{w}^{(t)})\|$). Computing these norms requires a time-consuming pass over the data. To make importance sampling practical, we propose RAIS-SGD.

## 4.1   Determining a robust sampling distribution

Like O-SGD, RAIS selects the $t$th minibatch by sampling indices from a discrete distribution $\mathbf{p}^{(t)}$. We denote the stochastic gradient by $\mathbf{g}_{\mathrm{R}}^{(t)}$, which takes the same form as $\mathbf{g}_{\mathrm{O}}^{(t)}$ in (4).

Let $v_i^* = \|\nabla f_i(\mathbf{w}^{(t)})\|$ and $\mathbf{v}^* = [v_1^*, v_2^*, \ldots, v_n^*]^T$. RAIS defines $\mathbf{p}^{(t)}$ by approximating $\mathbf{v}^*$. Naïve algorithms approximate $\mathbf{v}^*$ using a point estimate $\hat{\mathbf{v}}$. The sampling distribution becomes a multiple of $\hat{\mathbf{v}}$. [3], [4], and [6] propose algorithms based on similar point estimation strategies.

The drawback of the point estimation approach is extreme sensitivity to differences between $\hat{\mathbf{v}}$ and $\mathbf{v}^*$. For this reason, [3, 4, 6] incorporate additive smoothing. They introduce a hyperparameter, which we denote by $\delta$, and sample example $i$ with probability proportional to $\hat{v}_i + \delta$. This approach to robustness is unconvincing, however, since performance becomes critically dependent on a hyperparameter. Too small a $\delta$ risks divergence, while too large a value greatly limits the benefit of importance sampling.

Instead of a point estimate, RAIS approximates $\mathbf{v}^*$ with an uncertainty set $\mathcal{U}^{(t)} \subset \mathbb{R}_{\geq 0}^n$, which we expect contains (or nearly contains) $\mathbf{v}^*$. Given $\mathcal{U}^{(t)}$, RAIS defines $\mathbf{p}^{(t)}$ by minimizing the worst-case value of $\mathbb{E}[\|\mathbf{g}_{\mathrm{R}}^{(t)}\|^2]$ over all gradient norm possibilities in $\mathcal{U}^{(t)}$. Noting $\mathbb{E}[\|\mathbf{g}_{\mathrm{R}}^{(t)}\|^2] \propto \sum_i \frac{1}{p_i^{(t)}}(v_i^*)^2 + c$ for some $c \in \mathbb{R}$ (according to (5)), RAIS defines $\mathbf{p}^{(t)}$ as the solution to the following problem:

$$\mathbf{p}^{(t)} = \operatorname*{arginf}\left\{\max\left\{\textstyle\sum_{i=1}^n \frac{1}{p_i}v_i^2 \mid \mathbf{v} \in \mathcal{U}^{(t)}\right\} \mid \mathbf{p} \in \mathbb{R}_{>0}^n, \textstyle\sum_{i=1}^n p_i = 1\right\}. \qquad \text{(PRC)}$$

Such robust optimization problems are common for making decisions with data uncertainty [7].

It turns out (PRC) is straightforward to solve because the minimax theorem applies to (PRC) (we prove this in Appendix D.1, assuming our definition of $\mathcal{U}^{(t)}$ in §4.2). We first minimize over $\mathbf{p}$ by defining $p_i = v_i(\sum_{j=1}^n v_j)^{-1}$. Plugging this into (PRC)'s objective leads to the simplified problem

$$\mathbf{v}^{(t)} = \operatorname*{argmax}\left\{\left(\textstyle\sum_{i=1}^n v_i\right)^2 \mid \mathbf{v} \in \mathcal{U}^{(t)}\right\}. \qquad \text{(PRC')}$$

During each iteration $t$, RAIS solves (PRC'). After doing so, RAIS recovers the minimax optimal sampling distribution by defining $p_i^{(t)} \propto v_i^{(t)}$ for all training examples.

## 4.2   Modeling the uncertainty set

To define $\mathcal{U}^{(t)}$, RAIS uses features of SGD's state that are predictive of the true gradient norms. For each example $i$, we define a feature vector $\mathbf{s}_i^{(t)} \in \mathbb{R}_{\geq 0}^{d_{\mathrm{R}}}$. A useful feature for $\mathbf{s}_i^{(t)}$ is the gradient norm $\|\nabla f_i(\mathbf{w}^{(t')})\|$, where $t'$ is the most recent iteration for which $i \in \mathcal{M}^{(t')}$. Since RAIS-SGD computes $\nabla f_i(\mathbf{w}^{(t')})$ during iteration $t'$, constructing this feature during iteration $t$ should add little overhead.

Given $\mathbf{s}_i^{(t)}$ for all examples, RAIS defines the uncertainty set as an axis-aligned ellipsoid. Since $\mathbf{v}^* \geq 0$, RAIS also intersects this ellipsoid with the positive orthant. RAIS parameterizes this uncertainty set with two vectors, $\mathbf{c} \in \mathbb{R}_{\geq 0}^{d_{\mathrm{R}}}$ and $\mathbf{d} \in \mathbb{R}_{\geq 0}^{d_{\mathrm{R}}}$. These vectors map features $\mathbf{s}_{1:n}^{(t)}$ to parameters of the ellipsoid. Specifically, RAIS defines the uncertainty set as

$$\mathcal{U}_{\mathbf{cd}}^{(t)} = \left\{\mathbf{v} \in \mathbb{R}_{\geq 0}^n \,\big|\, \tfrac{1}{n}\sum_{i=1}^n Q_{\mathbf{cd}}(\mathbf{s}_i^{(t)}, v_i) \leq 1\right\}, \quad \text{where} \quad Q_{\mathbf{cd}}(\mathbf{s}, v) = \frac{(\langle \mathbf{c}, \mathbf{s}\rangle - v)^2}{\langle \mathbf{d}, \mathbf{s}\rangle}.$$

Here we denote the uncertainty set by $\mathcal{U}_{\mathbf{cd}}^{(t)}$ to emphasize the dependence of $\mathcal{U}^{(t)}$ on $\mathbf{c}$ and $\mathbf{d}$. With this definition of $\mathcal{U}_{\mathbf{cd}}^{(t)}$, (PRC') has a simple closed-form solution (proven in Appendix B):

**Proposition 4.1** (Solution to robust counterpart). *For all $i$, the solution to (PRC') satisfies*

$$v_i^{(t)} = \langle \mathbf{c}, \mathbf{s}_i^{(t)}\rangle + k\langle \mathbf{d}, \mathbf{s}_i^{(t)}\rangle, \quad \text{where} \quad k = \sqrt{n\big/ \textstyle\sum_{j=1}^n \langle \mathbf{d}, \mathbf{s}_j^{(t)}\rangle}.$$

If we consider $\langle \mathbf{c}, \mathbf{s}_i^{(t)}\rangle$ an estimate of $v_i^*$ and $\langle \mathbf{d}, \mathbf{s}_i^{(t)}\rangle$ a measure of uncertainty in this estimate, then Proposition 4.1 is quite interpretable. RAIS samples example $i$ with probability proportional to $\langle \mathbf{c}, \mathbf{s}_i^{(t)}\rangle + k\langle \mathbf{d}, \mathbf{s}_i^{(t)}\rangle$. The first term is the $v_i^*$ estimate, and the second term adds robustness to error.

### 4.3 Learning the uncertainty set

The uncertainty set parameters, $\mathbf{c}$ and $\mathbf{d}$, greatly influence the performance of RAIS. If $\mathcal{U}_{\mathbf{cd}}^{(t)}$ is a small region near $\mathbf{v}^*$, then RAIS's sampling distribution is similar to O-SGD's sampling distribution. If $\mathcal{U}_{\mathbf{cd}}^{(t)}$ is less representative of $\mathbf{v}^*$, the variance of the stochastic gradient could become much larger.

In order to make $\mathbb{E}[\|\mathbf{g}_{\mathrm{R}}^{(t)}\|^2]$ small but still ensure $\mathbf{v}^*$ likely lies in $\mathcal{U}_{\mathbf{cd}}^{(t)}$, RAIS adaptively defines $\mathbf{c}$ and $\mathbf{d}$. To do so, RAIS minimizes the size of $\mathcal{U}_{\mathbf{cd}}^{(t)}$ subject to a constraint that encourages $\mathbf{v}^* \in \mathcal{U}_{\mathbf{cd}}^{(t)}$:

$$\mathbf{c}, \mathbf{d} = \arginf\left\{\textstyle\sum_{i=1}^n \langle \mathbf{d}, \mathbf{s}_i^{(t)}\rangle \,\big|\, \mathbf{c}, \mathbf{d} \in \mathbb{R}_{\geq 0}^{d_{\mathrm{R}}}, \tfrac{1}{|\mathcal{D}|}\sum_{i=1}^{|\mathcal{D}|} \tilde{w}_i Q_{\mathbf{cd}}(\tilde{\mathbf{s}}_i, \tilde{v}_i) \leq 1\right\}. \tag{PT}$$

Here we have defined $\mathcal{U}_{\mathbf{cd}}^{(t)}$'s "size" as the sum of $\langle \mathbf{d}, \mathbf{s}_i^{(t)}\rangle$ values. The constraint that encourages $\mathbf{v}^* \in \mathcal{U}^{(t)}$ assumes weighted training data, $(\tilde{w}_i, \tilde{\mathbf{s}}_i, \tilde{v}_i)_{i=1}^{|\mathcal{D}|}$. RAIS must define this training set so that

$$\tfrac{1}{|\mathcal{D}|}\textstyle\sum_{i=1}^{|\mathcal{D}|} \tilde{w}_i Q_{\mathbf{cd}}(\tilde{\mathbf{s}}_i, \tilde{v}_i) \approx \tfrac{1}{n}\sum_{i=1}^n Q_{\mathbf{cd}}(\mathbf{s}_i^{(t)}, \|\nabla f_i(\mathbf{w}^{(t)})\|).$$

That is, for any $\mathbf{c}$ and $\mathbf{d}$, the mean of $Q_{\mathbf{cd}}(\tilde{\mathbf{s}}, \tilde{v}_i)$ over the weighted training set should approximately equal the mean of $Q_{\mathbf{cd}}(\mathbf{s}_i^{(t)}, v_i^*)$, which depends on current (unknown) gradient norms.

To achieve this, RAIS uses gradients from recent minibatches. For entry $j$ of the RAIS training set, RAIS considers an $i$ and $t'$ for which $i \in \mathcal{M}^{(t')}$ and $t' < t$. RAIS defines $\tilde{\mathbf{s}}_j = \mathbf{s}_i^{(t')}$, $\tilde{v}_j = \|\nabla f_i(\mathbf{w}^{(t')})\|$, and $\tilde{w}_j = (np_i^{(t')})^{-1}$. The justification for this choice is that the mean of $Q_{\mathbf{cd}}(\mathbf{s}_i^{(t)}, \|\nabla f_i(\mathbf{w}^{(t)})\|)$ over training examples tends to change gradually with $t$. Thus, the weighted mean over the RAIS training set approximates the mean of current $Q_{\mathbf{cd}}(\mathbf{s}_i^{(t)}, \|\nabla f_i(\mathbf{w}^{(t)})\|)$ values.

### 4.4 Approximating the gain ratio

In addition to the sampling distribution, RAIS must approximate the gain ratio in O-SGD. Define $\mathbf{g}_{\mathrm{R1}}^{(t)}$ as a stochastic gradient of the form (4) using minibatch size 1 and RAIS sampling. Define $\mathbf{g}_{\mathrm{U1}}^{(t)}$ in the same way but with uniform sampling. From (5), we can work out that the gain ratio satisfies

$$\mathbb{E}\left[\|\mathbf{g}_{\mathrm{U}}^{(t)}\|^2\right] \Big/ \mathbb{E}\left[\|\mathbf{g}_{\mathrm{R}}^{(t)}\|^2\right] = 1 + \tfrac{1}{|\mathcal{M}|}\left(\mathbb{E}[\|\mathbf{g}_{\mathrm{U1}}^{(t)}\|^2] - \mathbb{E}[\|\mathbf{g}_{\mathrm{R1}}^{(t)}\|^2]\right) \Big/ \mathbb{E}[\|\mathbf{g}_{\mathrm{R}}^{(t)}\|^2]. \tag{7}$$

To approximate the gain ratio, RAIS estimates the three moments on the right side of this equation. RAIS estimates $\mathbb{E}[\|\mathbf{g}_{\mathrm{R}}^{(t)}\|^2]$ using an exponential moving average of $\|\mathbf{g}_{\mathrm{R}}^{(t)}\|^2$ from recent iterations:

$$\mathbb{E}[\|\mathbf{g}_{\mathrm{R}}^{(t)}\|^2] \approx \alpha\left[\|\mathbf{g}_{\mathrm{R}}^{(t)}\|^2 + (1-\alpha)\|\mathbf{g}_{\mathrm{R}}^{(t-1)}\|^2 + (1-\alpha)^2\|\mathbf{g}_{\mathrm{R}}^{(t-2)}\|^2 + \dots\right].$$

**Algorithm 4.1** RAIS-SGD

---

**input** objective function $F$, minibatch size $|\mathcal{M}|$, learning rate schedule $\texttt{lr\_sched}(\cdot)$
**input** RAIS training set size $|\mathcal{D}|$, exponential smoothing parameter $\alpha$ for gain estimate
**initialize** $\mathbf{w}^{(1)} \in \mathbb{R}^d, \mathbf{c}, \mathbf{d} \in \mathbb{R}^{d_R}_{\geq 0}; \hat{t}^{(1)} \leftarrow 1; \texttt{r\_estimator} \leftarrow \texttt{GainEstimator}(\alpha)$
**for** $t = 1, 2, \ldots, T$ **do**
    $\mathbf{v}^{(t)} \leftarrow \operatorname{argmax}\left\{ \left(\sum_{i=1}^{n} v_i\right)^2 \mid \mathbf{v} \in \mathcal{U}^{(t)}_{\mathbf{cd}} \right\}$     *# see Proposition 4.1 for closed-form solution*
    $\mathbf{p}^{(t)} \leftarrow \mathbf{v}^{(t)}/\|\mathbf{v}^{(t)}\|_1$
    $\mathcal{M}^{(t)} \leftarrow \texttt{sample\_indices\_from\_distribution}(\mathbf{p}^{(t)}, \texttt{size} = |\mathcal{M}|)$
    $\mathbf{g}^{(t)}_R \leftarrow \frac{1}{|\mathcal{M}|} \sum_{i \in \mathcal{M}^{(t)}} \frac{1}{np^{(t)}_i} \nabla f_i(\mathbf{w}^{(t)}) + \lambda \mathbf{w}^{(t)}$
    $\texttt{r\_estimator.record\_gradient\_norms}(\|\mathbf{g}^{(t)}_R\|, (\|\nabla f_i(\mathbf{w}^{(t)})\|, p^{(t)}_i)_{i \in \mathcal{M}^{(t)}})$
    $\hat{r}^{(t)} \leftarrow \texttt{r\_estimator.estimate\_gain\_ratio}()$     *# see §4.4*
    $\eta^{(t)} \leftarrow \hat{r}^{(t)} \cdot \texttt{lr\_sched}(\hat{t}^{(t)})$
    $\mathbf{w}^{(t+1)} \leftarrow \mathbf{w}^{(t)} - \eta^{(t)} \mathbf{g}^{(t)}_R$
    $\hat{t}^{(t+1)} \leftarrow \hat{t}^{(t)} + \hat{r}^{(t)}$
    **if** $\operatorname{mod}(t, \lceil |\mathcal{D}|/|\mathcal{M}| \rceil) = 0$ **and** $t \geq (n + |\mathcal{D}|)/|\mathcal{M}|$ **then**
        $\mathbf{c}, \mathbf{d} \leftarrow \texttt{train\_uncertainty\_model}()$    *# see §4.2*
**return** $\mathbf{w}^{(T+1)}$

---

RAIS approximates $\mathbb{E}[\|\mathbf{g}^{(t)}_{R1}\|^2]$ and $\mathbb{E}[\|\mathbf{g}^{(t)}_{U1}\|^2]$ in a similar way. After computing gradients for minibatch $t$, RAIS estimates $\mathbb{E}[\|\mathbf{g}^{(t)}_{R1}\|^2]$ and $\mathbb{E}[\|\mathbf{g}^{(t)}_{U1}\|^2]$ using appropriately weighted averages of $\|\nabla f_i(\mathbf{w}^{(t)})\|^2$ for each $i \in \mathcal{M}^{(t)}$ (for $\mathbb{E}[\|\mathbf{g}^{(t)}_{R1}\|^2]$, RAIS weights terms by $(np^{(t)}_i)^{-2}$; for $\mathbb{E}[\|\mathbf{g}^{(t)}_{U1}\|^2]$, RAIS weights terms by $(np^{(t)}_i)^{-1}$). Using the same exponential averaging parameter $\alpha$, RAIS averages these estimates from minibatch $t$ with estimates from prior iterations.

RAIS approximates the gain ratio by plugging these moment estimates into (7). We denote the result by $\hat{r}^{(t)}$. Analogous to O-SGD, RAIS uses learning rate $\eta^{(t)} = \hat{r}^{(t)} \texttt{lr\_sched}(\hat{t}^{(t)})$, where $\hat{t}^{(t)}$ is the effective iteration number: $\hat{t}^{(t)} = \sum_{t'=1}^{t} \hat{r}^{(t'-1)}$. Here we also define the edge case $\hat{r}^{(0)} = 1$.

## 4.5 Practical considerations

Algorithm 4.1 summarizes our RAIS-SGD algorithm. We next discuss important practical details.

**Solving (PT)** While computing $\mathbf{p}^{(t)}$ requires a small number of length $n$ operations (see Proposition 4.1), learning the uncertainty set parameters requires more computation. For this reason, RAIS should not solve (PT) during every iteration. Our implementation solves (PT) asynchronously after every $\lceil |\mathcal{D}|/|\mathcal{M}| \rceil$ minibatches, with updates to $\mathbf{w}^{(t)}$ continuing during the process. We describe our algorithm for solving (PT) in Appendix D.2. Since our features $\mathbf{s}^{(t)}_{1:n}$ depend on past minibatch updates, we do not use RAIS for the first epoch of training—instead we sample examples sequentially.

**Compatibility with common tricks** RAIS combines nicely with standard training tricks for deep learning. With no change, we find RAIS works well with momentum [8, 9]. Incorporating data augmentation, dropout [10], or batch normalization [11] adds variance to the model's outputs and gradient norms. RAIS elegantly compensates for such inconsistency by learning a larger uncertainty set. Since the importance sampling distribution changes over time, we find it important to compute weighted batch statistics when using RAIS with batch normalization. That is, when computing normalization statistics during training, we weight contributions from each example by $(np^{(t)}_i)^{-1}$.

**Protecting against outliers** In some cases—typically when the gain ratio is very large—we find $Q_{\mathbf{cd}}(\mathbf{s}^{(t)}_i, v^*_i)$ can be quite small for most examples yet large for a small set of outliers. Typically we find RAIS does not require special treatment of such outliers. Even so, it is reasonable to protect against outliers, so that an example with extremely large $Q_{\mathbf{cd}}(\mathbf{s}^{(t)}_i, v^*_i)$ cannot greatly increase the stochastic gradient's variance. To achieve this, we use gradient clipping, and RAIS provides a natural

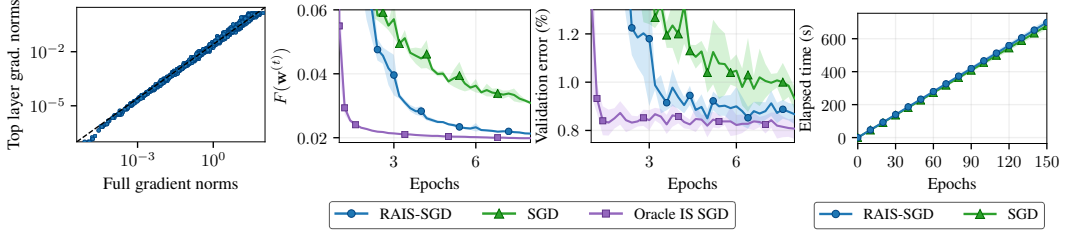

**Figure 2: Supplemental plots.** *Left:* Visualization of top-layer gradient norm approximation. The model is an 18 layer ResNet after 30 epochs of training on CIFAR-10. *Middle:* Oracle importance sampling results for MNIST and LeNet model. *Right:* RAIS time overhead for rot-MNIST.

way of doing so. We define an "outlier" as any example for which $Q_{\mathbf{cd}}(\mathbf{s}_i^{(t)}, v_i^*)$ exceeds a threshold $\tau$. For each outlier $i$, we temporarily scale $f_i$ during iteration $t$ until $Q_{\mathbf{cd}}(\mathbf{s}_i^{(t)}, \|\nabla f_i(\mathbf{w}^{(t)})\|) = \tau$. In practice, we use $\tau = 100$; the fraction of outliers is often zero and rarely exceeds $0.1\%$.

**Approximating per-example gradient norms**  To train the uncertainty set, RAIS computes $\|\nabla f_i(\mathbf{w}^{(t)})\|$ for each example in each minibatch. Unfortunately, existing software tools do not provide efficient access to per-example gradient norms. Instead, libraries are optimized for aggregating gradients over minibatches. Thus, to make RAIS practical, we must approximate the gradient norms. We do so by replacing $\|\nabla f_i(\mathbf{w}^{(t)})\|$ with the norm of only the loss layer's gradient (with respect to this layer's inputs). These values correlate strongly, since the loss layer begins the backpropagation chain for computing $\nabla f_i(\mathbf{w}^{(t)})$. We show this empirically in Figure 2 (left), and we include additional plots in Appendix E.1. We note this approximation may not work well for all models.

## 5   Relation to prior work

Prior strategies also consider importance sampling for speeding up deep learning. [3] proposes distributing the computation of sampling probabilities. In parallel with regular training, [4] trains a miniature neural network to predict importance values. [5] approximates importance values using additional forward passes. [12] and [13] apply importance sampling to deep reinforcement learning. With the exception of [5] (which requires considerable time to compute importance values), these prior algorithms are sensitive to errors in importance value estimates. For this reason, all require critical smoothing hyperparameters to converge. In contrast, RAIS elegantly compensates for approximation error by choosing a sampling distribution that is minimax optimal with respect to an uncertainty set. Since RAIS adaptively trains this uncertainty set, RAIS does not require hyperparameter tuning.

Researchers have also considered other ways to prioritize training examples for deep learning. [14] considers examples in order of increasing difficulty. Other researchers prioritize challenging training examples [15, 16]. And yet others prioritize examples closest to the model's decision boundary [17]. Unlike RAIS, the primary goal of these approaches is improved model performance, not optimization efficiency. Importance sampling may work well in conjunction with these strategies.

There also exist ideas for sampling minibatches non-uniformly outside the context of deep learning. [18, 19] consider sampling diverse minibatches via repulsive point processes. Another strategy uses side information, such as class labels, for approximate importance sampling [6]. By choosing appropriate features for the uncertainty set, RAIS can use side information in the same way.

In the convex setting, there are several importance sampling strategies for SGD with theoretical guarantees. This includes [1] and [2], which sample training examples proportional to Lipschitz constants. Leverage score sampling uses a closely related concept for matrix approximation algorithms [20, 21]. For more general convex problems, some adaptive sampling strategies include [22] and [23].

## 6   Empirical comparisons

In this section, we demonstrate how RAIS performs in practice. We consider the very popular task of training a convolutional neural network to classify images.

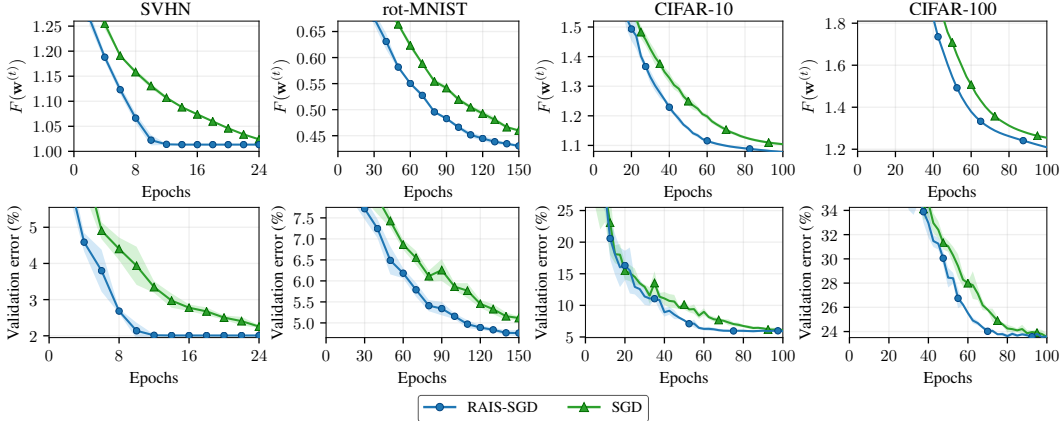

Figure 3: **Learning curve comparison.** RAIS consistently outperforms SGD with uniform sampling, both in terms of objective value and generalization performance. Curves show the mean of five trials with varying random seeds. Filled areas signify $\pm 1.96$ times standard error of the mean.

We first train a LeNet-5 model [24] on the MNIST digits dataset. The model's small size makes it possible to compare with O-SGD. We use learning rate $\eta^{(t)} = 3.4/\sqrt{100 + t}$, L2 penalty $\lambda = 2.5 \times 10^{-4}$, and batch size 32—the parameters are chosen so that SGD performs well. We do not use momentum or data augmentation. Figure 2(middle) includes the results of this experiment. Oracle sampling significantly outperforms RAIS, and RAIS significantly outperforms uniform sampling.

For our remaining comparisons, we consider street view house numbers [25], rotated MNIST [26], and CIFAR tiny image [27] datasets. For rot-MNIST, we train a 7 layer CNN with 20 channels per layer—a strong baseline from [28]. Otherwise, we train an 18 layer ResNet preactivation model [29]. CIFAR-100 contains 100 classes, while the other problems contain 10. The number of training examples is $6.0 \times 10^5$ for SVHN, $1.2 \times 10^4$ for rot-MNIST, and $5.0 \times 10^4$ for the CIFAR problems.

We follow standard training procedures to attain good generalization performance. We use batch normalization and standard momentum of 0.9. For rot-MNIST, we follow [28], augmenting data with random rotations and training with dropout. For the CIFAR problems, we augment the training set with random horizontal reflections and random crops (pad to 40x40 pixels; crop to 32x32).

We train the SVHN model with batch size 64 and the remaining models with $|\mathcal{M}| = 128$. For each problem, we approximately optimize $\lambda$ and the learning rate schedule in order to achieve good validation performance with SGD at the end of training. The learning rate schedule decreases by a fixed fraction after each epoch ($n/|\mathcal{M}|$ iterations). This fraction is 0.8 for SVHN, 0.972 for rot-MNIST, 0.96 for CIFAR-10, and 0.96 for CIFAR-100. The initial learning rates are 0.15, 0.09, 0.08, and 0.1, respectively. We use $\lambda = 3 \times 10^{-3}$ for rot-MNIST and $\lambda = 5 \times 10^{-4}$ otherwise.

For RAIS-SGD, we use $|\mathcal{D}| = 2 \times 10^4$ training examples to learn $\mathbf{c}$ and $\mathbf{d}$ and $\alpha = 0.01$ to estimate $\hat{r}^{(t)}$. The performance of RAIS varies little with these parameters, since they only determine the number of minibatches to consider when training the uncertainty set and estimating the gain ratio. For the uncertainty set features, we use simple moving averages of the most recently computed gradient norms for each example. We use moving averages of different lengths—1, 2, 4, 8, and 16. For lengths of at least four, we also include the variance and standard deviation of these prior gradient norm values. We also incorporate a bias feature as well as the magnitude of the random crop offset.

We compare training curves of RAIS-SGD and SGD in Figure 3. Notice that RAIS-SGD consistently outperforms SGD. The relative speed-up ranges from approximately 20% for the CIFAR-100 problem to more than 2x for the SVHN problem. Due to varying machine loads, we plot results in terms of epochs (not wall time), but RAIS introduces very little time overhead. For example, Figure 2(right) includes time overhead results for the rot-MNIST comparison, which we ran on an isolated machine.

Figure 4 provides additional details of these results. In the figure's first row, we see the speed-up in terms of the gain ratio (the blue curve averages the value $(\hat{r}^{(t)} - 1) \cdot 100\%$ over consecutive epochs).

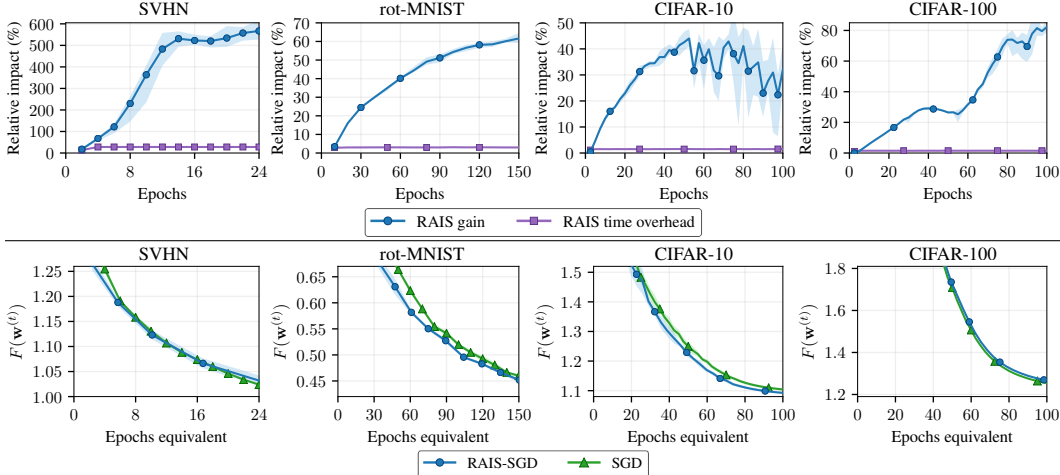

Figure 4: **RAIS speed-up and alignment of epochs equivalent.** *Above:* Blue shows increase in optimization speed due to RAIS, as measured by estimated gain ratio; purple indicates time overhead due to RAIS. Overhead is small compared to speed-up. *Below:* Objective value vs. epochs equivalent. For RAIS, epochs equivalent equals $\frac{|\mathcal{M}|}{n}\hat{t}^{(t)}$. The closely aligned curves suggest (i) RAIS-SGD is a suitable drop-in replacement for SGD, and (ii) the gain ratio correctly approximates speed-up.

The gain ratio tends to increase as training progresses, implying RAIS is most useful during later stages of training. We also plot the relative wall time overhead for RAIS, which again is very small.

In the second row of Figure 4, we compare RAIS-SGD and SGD in terms of *epochs equivalent*—the number of epochs measured in terms of effective iterations. Interestingly, the curves align closely. This alignment confirms that our learning rate adjustment is reasonable, as it results in a suitable drop-in replacement for SGD. This result contrasts starkly with [3], for example, in which case generalization performance differs significantly for the importance sampling and standard algorithms.

Table 1 concludes these comparisons with a summary of results:

Table 1: **Quantities upon training completion.**

| Dataset | Algorithm | $F(\mathbf{w}^{(t)})$ | Val. error | Val. loss | Epochs equivalent |
|---|---|---|---|---|---|
| SVHN | RAIS-SGD | **1.01** | **0.0201** | **0.121** | **114** |
| | SGD | 1.02 | 0.0226 | **0.121** | 24.0 |
| rot-MNIST | RAIS-SGD | **0.431** | **0.0476** | **0.149** | **214** |
| | SGD | 0.460 | 0.0512 | 0.161 | 150. |
| CIFAR-10 | RAIS-SGD | **1.08** | **0.0590** | **0.256** | **130.** |
| | SGD | 1.10 | 0.0607 | 0.277 | 100. |
| CIFAR-100 | RAIS-SGD | **1.21** | **0.236** | **0.962** | **138** |
| | SGD | 1.25 | **0.236** | 0.989 | 100. |

## 7 Discussion

We proposed a relatively simple and very practical importance sampling procedure for speeding up the training of deep models. By using robust optimization to define the sampling distribution, RAIS depends minimally on user-specified parameters. Additionally, RAIS introduces little computational overhead and combines nicely with standard training strategies. All together, RAIS is a promising approach with minimal downside and potential for large improvements in training speed.

## Acknowledgements

We thank Marco Tulio Ribeiro, Tianqi Chen, Maryam Fazel, Sham Kakade, and Ali Shojaie for helpful discussion and feedback. This work was supported by PECASE N00014-13-1-0023.

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
