[Supplementary Material · rais-with-appendices.pdf]

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

$\quad \mathbf{v}^{(t)} \leftarrow \operatorname{argmax}\big\{ \big(\sum_{i=1}^n v_i\big)^2 \mid \mathbf{v} \in \mathcal{U}_{\mathbf{cd}}^{(t)} \big\}$ $\quad$ *# see Proposition 4.1 for closed-form solution*
$\quad \mathbf{p}^{(t)} \leftarrow \mathbf{v}^{(t)}/\|\mathbf{v}^{(t)}\|_1$
$\quad \mathcal{M}^{(t)} \leftarrow \texttt{sample\_indices\_from\_distribution}(\mathbf{p}^{(t)}, \texttt{size} = |\mathcal{M}|)$
$\quad \mathbf{g}_{\mathrm{R}}^{(t)} \leftarrow \frac{1}{|\mathcal{M}|} \sum_{i \in \mathcal{M}^{(t)}} \frac{1}{np_i^{(t)}} \nabla f_i(\mathbf{w}^{(t)}) + \lambda \mathbf{w}^{(t)}$
$\quad \texttt{r\_estimator.record\_gradient\_norms}(\|\mathbf{g}_{\mathrm{R}}^{(t)}\|, (\|\nabla f_i(\mathbf{w}^{(t)})\|, p_i^{(t)})_{i \in \mathcal{M}^{(t)}})$
$\quad \hat{r}^{(t)} \leftarrow \texttt{r\_estimator.estimate\_gain\_ratio}()$ $\quad$ *# see §4.4*
$\quad \eta^{(t)} \leftarrow \hat{r}^{(t)} \cdot \texttt{lr\_sched}(\hat{t}^{(t)})$
$\quad \mathbf{w}^{(t+1)} \leftarrow \mathbf{w}^{(t)} - \eta^{(t)} \mathbf{g}_{\mathrm{R}}^{(t)}$
$\quad \hat{t}^{(t+1)} \leftarrow \hat{t}^{(t)} + \hat{r}^{(t)}$
$\quad$ **if** $\operatorname{mod}(t, \lceil |\mathcal{D}|/|\mathcal{M}| \rceil) = 0$ **and** $t \geq (n + |\mathcal{D}|)/|\mathcal{M}|$ **then**
$\quad\quad \mathbf{c}, \mathbf{d} \leftarrow \texttt{train\_uncertainty\_model}()$ $\quad$ *# see §4.2*
**return** $\mathbf{w}^{(T+1)}$

---

RAIS approximates $\mathbb{E}[\|\mathbf{g}_{\mathrm{R1}}^{(t)}\|^2]$ and $\mathbb{E}[\|\mathbf{g}_{\mathrm{U1}}^{(t)}\|^2]$ in a similar way. After computing gradients for minibatch $t$, RAIS estimates $\mathbb{E}[\|\mathbf{g}_{\mathrm{R1}}^{(t)}\|^2]$ and $\mathbb{E}[\|\mathbf{g}_{\mathrm{U1}}^{(t)}\|^2]$ using appropriately weighted averages of $\|\nabla f_i(\mathbf{w}^{(t)})\|^2$ for each $i \in \mathcal{M}^{(t)}$ (for $\mathbb{E}[\|\mathbf{g}_{\mathrm{R1}}^{(t)}\|^2]$, RAIS weights terms by $(np_i^{(t)})^{-2}$; for $\mathbb{E}[\|\mathbf{g}_{\mathrm{U1}}^{(t)}\|^2]$, RAIS weights terms by $(np_i^{(t)})^{-1}$).

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

# A   Proof of Proposition 3.1

This appendix shows work for deriving (5) and the oracle importance sampling distribution.

## A.1   Equation for gradient second moment

To derive (5), first define

$$\nabla f_{\mathcal{M}^{(t)}}(\mathbf{w}^{(t)}) = \mathbf{g}_O^{(t)} - \lambda \mathbf{w}^{(t)} .$$

We have

$$
\begin{aligned}
\mathbb{E}\left[\|\mathbf{g}_O^{(t)}\|^2\right] &= \mathbb{E}\left[\|\mathbf{g}_O^{(t)} - \nabla \bar{f}(\mathbf{w}^{(t)}) + \nabla \bar{f}(\mathbf{w}^{(t)})\|^2\right] \\
&= \mathbb{E}\left[\|\nabla f_{\mathcal{M}^{(t)}}(\mathbf{w}^{(t)}) - \bar{f}(\mathbf{w}^{(t)}) + \nabla F(\mathbf{w}^{(t)})\|^2\right] \\
&= \mathbb{E}\left[\|\nabla f_{\mathcal{M}^{(t)}}(\mathbf{w}^{(t)}) - \bar{f}(\mathbf{w}^{(t)})\|^2\right] + \|\nabla F(\mathbf{w}^{(t)})\|^2 . \qquad (8)
\end{aligned}
$$

Above, we used the fact that $\mathbb{E}\left[\nabla f_{\mathcal{M}^{(t)}}(\mathbf{w}^{(t)})\right] = \bar{f}(\mathbf{w}^{(t)})$.

Continuing, we have

$$
\begin{aligned}
\mathbb{E}\left[\|\nabla f_{\mathcal{M}^{(t)}}(\mathbf{w}^{(t)}) - \bar{f}(\mathbf{w}^{(t)})\|^2\right] &= \mathbb{E}\left[\left\|\frac{1}{|\mathcal{M}|}\sum_{i \in \mathcal{M}^{(t)}} \frac{1}{n p_i^{(t)}} \nabla f_i(\mathbf{w}^{(t)}) - \bar{f}(\mathbf{w}^{(t)})\right\|^2\right] \\
&= \frac{1}{|\mathcal{M}|^2}\mathbb{E}\left[\left\|\sum_{i \in \mathcal{M}^{(t)}}\left(\frac{1}{n p_i^{(t)}}\nabla f_i(\mathbf{w}^{(t)}) - \bar{f}(\mathbf{w}^{(t)})\right)\right\|^2\right] \\
&= \frac{1}{|\mathcal{M}|}\mathbb{E}\left[\left\|\frac{1}{n p_i^{(t)}}\nabla f_i(\mathbf{w}^{(t)}) - \bar{f}(\mathbf{w}^{(t)})\right\|^2\right] \\
&= \frac{1}{|\mathcal{M}|}\left(\mathbb{E}\left[\left\|\frac{1}{n p_i^{(t)}}\nabla f_i(\mathbf{w}^{(t)})\right\|^2\right] - \left\|\bar{f}(\mathbf{w}^{(t)})\right\|^2\right) .
\end{aligned}
$$

Combining with (8) leads to the result:

$$\mathbb{E}\left[\|\mathbf{g}_O^{(t)}\|^2\right] = \frac{1}{|\mathcal{M}|n^2}\sum_{i=1}^{n}\frac{1}{p_i^{(t)}}\left\|\nabla f_i(\mathbf{w}^{(t)})\right\|^2 - \frac{1}{|\mathcal{M}|}\left\|\bar{f}(\mathbf{w}^{(t)})^2\right\| + \left\|\nabla F(\mathbf{w}^{(t)})\right\|^2 .$$

## A.2   Finding the optimal sampling distribution

We want to find the distribution $\mathbf{p}$ that minimizes $\sum_{i=1}^{n} p_i^{-1}(v_i^*)^2$, where $v_i^* = \left\|\nabla f_i(\mathbf{w}^{(t)})\right\|$. From Jensen's inequality, it follows that

$$\sum_{i=1}^{n} p_i^{-1}(v_i^*)^2 = \sum_{i=1}^{n} p_i\left(\frac{v_i^*}{p_i}\right)^2 \geq \left(\sum_{i=1}^{n} v_i^*\right)^2 .$$

When $p_i = v_i^*/\sum_j v_j^*$, the inequality is satisfied with equality. Thus, this choice for the sampling distribution is optimal.

# B  Proof of Proposition 4.1

*Proof.* Since $\mathbf{v} \geq 0$, minimizing $(\sum_i v_i)^2$ is equivalent to minimizing $\sum_i v_i$. For some $\nu \geq 0$, the maximizer of $\sum_i v_i$ subject to $\mathbf{v} \in \mathcal{U}_{\mathbf{cd}}^{(t)}$ satisfies

$$\nabla \sum\nolimits_i v_i = \nu \nabla \tfrac{1}{n} \sum\nolimits_i Q_{\mathbf{cd}}(v_i) \quad \text{and} \quad \nu(\tfrac{1}{n} \sum\nolimits_i Q_{\mathbf{cd}}(v_i) - 1) = 0 \,.$$

The solution given by the proposition satisfies these KKT conditions when $\nu = \frac{n}{2k}$. $\qquad\square$

# C  Elaboration on (7)

We need to show

$$\mathbb{E}\left[\|\mathbf{g}_{\mathrm{U}}^{(t)}\|^2\right] = \mathbb{E}\left[\|\mathbf{g}_{\mathrm{R}}^{(t)}\|^2\right] + \tfrac{1}{|\mathcal{M}|}\left(\mathbb{E}[\|\mathbf{g}_{\mathrm{U}1}^{(t)}\|^2] - \mathbb{E}[\|\mathbf{g}_{\mathrm{R}1}^{(t)}\|^2]\right) \,.$$

This follows algebraically from (5). Let $c_{|\mathcal{M}|} = \|\nabla F(\mathbf{w}^{(t)})\|^2 - \frac{1}{|\mathcal{M}|}\|\nabla \bar{f}(\mathbf{w}^{(t)})\|^2$, and let $c_1 = \|\nabla F(\mathbf{w}^{(t)})\|^2 - \|\nabla \bar{f}(\mathbf{w}^{(t)})\|^2$. From (5), we have

$$\mathbb{E}\left[\|\mathbf{g}_{\mathrm{U}}^{(t)}\|^2\right] = \tfrac{1}{n|\mathcal{M}|} \sum_{i=1}^{n} \|\nabla f_i(\mathbf{w}^{(t)})\|^2 + c_{|\mathcal{M}|}$$

$$\mathbb{E}\left[\|\mathbf{g}_{\mathrm{R}}^{(t)}\|^2\right] = \tfrac{1}{n^2|\mathcal{M}|} \sum_{i=1}^{n} \tfrac{1}{p_i^{(t)}} \|\nabla f_i(\mathbf{w}^{(t)})\|^2 + c_{|\mathcal{M}|}$$

$$\mathbb{E}\left[\|\mathbf{g}_{\mathrm{U}1}^{(t)}\|^2\right] = \tfrac{1}{n} \sum_{i=1}^{n} \|\nabla f_i(\mathbf{w}^{(t)})\|^2 + c_1$$

$$\mathbb{E}\left[\|\mathbf{g}_{\mathrm{R}1}^{(t)}\|^2\right] = \tfrac{1}{n^2} \sum_{i=1}^{n} \tfrac{1}{p_i^{(t)}} \|\nabla f_i(\mathbf{w}^{(t)})\|^2 + c_1 \,.$$

# D  Details of solving (PRC) and (PT)

This appendix provides details of solving the optimization problems in RIAS—specifically solving the robust counterpart and training the uncertainty set.

## D.1  Justification that the minimax theorem applies to (PRC)

We use the change of variables $u_i = v_i^2$. Define

$$\tilde{\mathcal{U}}^{(t)} = \left\{ \mathbf{u} \in \mathbb{R}_{\geq 0}^n \mid \tfrac{1}{n} \sum_{i=1}^{n} \tfrac{(\langle \mathbf{c}, \mathbf{s}_i^{(t)} \rangle - \sqrt{u_i})^2}{\langle \mathbf{d}, \mathbf{s}_i^{(t)} \rangle} \leq 1 \right\} \,.$$

Note (i) $\mathbf{u} \in \tilde{\mathcal{U}}^{(t)} \Leftrightarrow \mathbf{v} \in \mathcal{U}^{(t)}$, and (ii) $\tilde{\mathcal{U}}^{(t)}$ is compact and convex, since $\langle \mathbf{c}, \mathbf{s}_i^{(t)} \rangle \geq 0$ and $\langle \mathbf{d}, \mathbf{s}_i^{(t)} \rangle > 0$ for all $i \in [n]$. Let $\mathcal{P} = \{\mathbf{p} \in \mathbb{R}_{>0}^n \mid \sum_{i=1}^{n} p_i = 1\}$. The robust counterpart is

$$\inf_{\mathbf{p} \in \mathcal{P}} \max_{\mathbf{u} \in \tilde{\mathcal{U}}^{(t)}} \sum_{i=1}^{n} \tfrac{1}{np_i} u_i \,.$$

This objective is separately concave in $\mathbf{u}$ and convex in $\mathbf{p}$. We have shown the conditions for the minimax theorem. Thus, to solve this problem, we can first optimize over $\mathbf{p}$ by setting $p_i \propto \sqrt{u_i}$. Afterward, we can optimize over $\mathbf{u}$.

## D.2 Approach to solving (PT)

We need to solve

$$\underset{\mathbf{c},\mathbf{d}\geq 0}{\text{minimize}} \quad \sum_{i=1}^{n}\langle\mathbf{d},\mathbf{s}_i^{(t)}\rangle$$
$$\text{s.t.} \qquad \frac{1}{|\mathcal{D}|}\sum_{i=1}^{|\mathcal{D}|}\tilde{w}_i\frac{(\langle\mathbf{c},\tilde{\mathbf{s}}_i\rangle-\tilde{v}_i)^2}{\langle\mathbf{d},\tilde{\mathbf{s}}_i\rangle}\leq 1\,.$$

We reduce the problem to the unconstrained problem

$$\underset{\mathbf{c},\mathbf{d}\geq 0}{\text{minimize}}\ \frac{1}{n}\sum_{i=1}^{n}\langle\mathbf{d},\mathbf{s}_i^{(t)}\rangle + \frac{1}{|\mathcal{D}|}\sum_{i=1}^{|\mathcal{D}|}\tilde{w}_i\frac{(\langle\mathbf{c},\tilde{\mathbf{s}}_i\rangle-\tilde{v}_i)^2}{\langle\mathbf{d},\tilde{\mathbf{s}}_i\rangle}\,.$$

These problems have the same solution up to a scaling of $\mathbf{d}$. After solving the second problem, we can recover the solution to the first problem by scaling $\mathbf{d}$ in order to satisfy the first problem's constraint with equality.

We minimize this objective using alternating minimization. Each separate update to $\mathbf{c}$ and $\mathbf{d}$ is a Newton step that we compute with a nonnegative least squares solver.

# E  Additional empirical results

This appendix includes additional empirical results that we did not include in the main text due to space constraints.

## E.1  Empirical justification of gradient norm approximation

Figure 2(left) includes a scatter plot that justifies our approximate measurements of per-example gradient norms. Here we include additional plots to further support this approximation.

For the following plots, we consider the CIFAR-10 ResNet model after varying amounts of training:

The next set of plots shows similar results for additional models:

Since only the relative gradient magnitudes matter when determining the importance sampling distribution, the approximation is quite effective in these cases.

## E.2  Validation error vs. epochs equivalent

In Figure 4, we plot training objective vs. epochs equivalent. This empirically justifies our strategy for adapting the learning rate.

The following plots show that the curves also align closely when considering validation error:

## E.3  Comparisons using alternative learning rate schedule

In §6, our learning rate schedule decreased $\eta^{(t)}$ by a multiplicative factor after each epoch. For each problem, we optimized the learning rate parameters so that SGD performed well.

There is another learning rate schedule that is common for training ResNet models. Following [29], we can initialize the learning rate at $0.1$. We then decrease the learning rate to $0.01$ after training is $50\%$ completed. Finally, after training is $75\%$ completed, we drop the learning rate to $0.001$.

When using this learning rate schedule, we have the following results for the ResNet models:

We can also consider the alignment of curves when plotted in terms of epochs equivalent:

The final set of plots compares the speed-up and time overhead due to RAIS:

## E.4 Importance of rescaling the learning rate

For our final comparison, we show plots that demonstrate the importance of rescaling the learning rate when using RAIS. We train a LeNet-5 model [24] on the MNIST digit recognition dataset. We use the learning rate schedule $\eta^{(t)} = \eta_0 / \sqrt{100/(100 + t)}$ with regularization penalty $\lambda = 2.5 \times 10^{-4}$. We do not use momentum or data augmentation.

We compare RAIS-SGD with standard SGD as well as a control RAIS-SGD algorithm for which we do not adapt the learning rate (i.e., we define $\hat{r}^{(t)} = 1$ for all $t$). In the top row, we include results for $\eta_0 = 1.0$, which is a large choice of learning rate. (When setting $\eta_0 = 1.1$, for example, we found that SGD occasionally did not converge.) In the second row, we include results for $\eta_0 = 0.25$.

In the large learning rate scenario, we see that RAIS improves training times, regardless of whether we rescale the learning rate. In the small learning rate scenario, however, adapting the learning rate is crucial for obtaining significant speed-ups.