[Reviews · NeurIPS 2018]

Reviewer 1



The authors present a method to make Importance Sampling SGD more robust. There are a few difficulties with the vanilla algorithm, and one of them is the instability of the importance weights. The authors proposed to address this by introducing a method that estimates the quantities involved. The authors briefly refer to the earlier work by Alain et al (2016) in their introduction, but make no effort to describe any of that work which is similar to theirs. The authors present their “Oracle SGD” algorithm using the same kind of language as would be used to introduce a novel idea. The whole paper takes great pains to devise a novel method that makes the Importance Sampling SGD more robust, but at the same time they casually claim that it’s quite reasonable to use the gradient norm of the last layer as approximation to the gradient norm over the whole model. This is stated entirely without reference, and I suspect that it is not even true as a general fact (while it might be true for certain models at a certain time of training on certain data). Otherwise, the “exploding gradient” problem would not be a thing. They could have at least provided a sanity check on their own data and model just to make sure that they weren’t completely wrong about this. This can be done with a batch size 1, at an increased cost for sure, but it doesn’t need to be done more than a few times. If it is indeed true, it does make it easier to apply their method to any model because it’s not so hard to compute the gradient norms on the final layers. Figure 1 is especially nice and intuitive. Figure 3 seems to suggest that epochs run faster, but I believe that they are comparing to evaluating importance samples on the whole training set. This is an interesting quantity, but it seems like it would be something that advocates of Importance Sampling SGD would not actually recommend simply due to the fact that it would scale badly with the size of the training set. It would be interesting to compare RAIS-SGD’s speed with other reasonable methods that, for example, run Importance Sampling SGD on only a fifth of the training set (corresponding to a 500% speed increase but possibly not an increase in training quality).

Reviewer 2



At least in the purposes and goals, the proposed method or better this work seems similar to Ö. D. Akyıldız, I. P. Marino, J. Miguez. Adaptive noisy importance sampling for stochastic optimization, IEEE CAMSAP 2017. However, the point is that, in my opinion, the paper is very confused. The presentation must be improved. It is even difficult to see where the importance sampling in Section 3.2. Secondly, several recent references of importance sampling are missed. Therefore, this paper is not ready for publication.

Reviewer 3



The paper proposes a robust way to approximate optimal importance sampling weights to reduce the variance of stochastic gradient for training machine learning models. I think the idea (and implementation) is definitely valuable for the community and deserves a publication, but improving experiments can make the paper much more useful and insightful. In particular, I believe that 1) the exact O-SGD baseline (sampling without approximations) should be included in the plots at least for small-scale experiments; 2) there should be not only epoch plots, but the time plots as well to directly see the overhead of the sampling scheme; 3) there should be oblation studies for the factors you think make your method particularly effective (the ones listed in sec 6.2: large dataset size, lack of random data augmentation; etc). Also, you mention that you use only the loss layer of the networks to compute the gradient norms, which I see as a major approximation. Consider using the method from [1] at least as a baseline if the overhead is too large to use it on practice. Again, the paper is really good, but I would love to see it getting even better before the publication. Minor points: Would be nice to have pointer to appendix whenever you included the proof in where. E.g. now it looks like you consider proof of (4) too trivial to write down and I spent some time to rederive it only to find later that it’s actually proven in the appendix. The introduction of the dataset D on line 109 is confusing because it appears out of nowhere and when get explained slowly. Also, I read this part many times and still not sure how do you choose objects for D. I don’t get equation (6): how does it follow from (4) and what is g_{U1} and g_{R1}? Figure 6.2 referenced on line 228 doesn’t seem to exist Since you mentioned generalization as something you pay attention to (line 237), it would be nice to have test errors as well (in contrast to only having validation errors, since you used validation set for hyperparameters tuning). BTW, did you find optimal hyperparameters for SGD and for your method independently? [1] Goodfellow, Ian. "Efficient per-example gradient computations." arXiv preprint arXiv:1510.01799 (2015). UPD: thanks for addressing my concerns, I'm now even more willing to accept the paper.